# Emerging Evidence and Treatment Perspectives from Randomized Clinical Trials in Systemic Sclerosis: Focus on Interstitial Lung Disease

**DOI:** 10.3390/biomedicines10020504

**Published:** 2022-02-21

**Authors:** Caterina Oriana Aragona, Antonio Giovanni Versace, Carmelo Ioppolo, Daniela La Rosa, Rita Lauro, Maria Concetta Tringali, Simona Tomeo, Guido Ferlazzo, William Neal Roberts, Alessandra Bitto, Natasha Irrera, Gianluca Bagnato

**Affiliations:** 1Department of Clinical and Experimental Medicine, University of Messina, 98125 Messina, Italy; oriana.aragona@gmail.com (C.O.A.); antoniogiovanni.versace@polime.it (A.G.V.); carmelo.ioppolo@polime.it (C.I.); daniela.larosa@polime.it (D.L.R.); irritalauro@gmail.com (R.L.); mariaconcetta.tringali@polime.it (M.C.T.); simona.tomeo@polime.it (S.T.); abitto@unime.it (A.B.); nirrera@unime.it (N.I.); 2Department of Human Pathology “G. Barresi”, University of Messina, 98125 Messina, Italy; guido.ferlazzo@unime.it; 3Department of Medicine, University of Kentucky, Lexington, KY 40506, USA; neal.roberts@uky.edu

**Keywords:** systemic sclerosis, interstitial lung disease, clinical trial

## Abstract

Systemic sclerosis (SSc) is a complex rare autoimmune disease with heterogeneous clinical manifestations. Currently, interstitial lung disease (ILD) and cardiac involvement (including pulmonary arterial hypertension) are recognized as the leading causes of SSc-associated mortality. New molecular targets have been discovered and phase II and phase III clinical trials published in the last 5 years on SSc-ILD will be discussed in this review. Details on the study design; the drug tested and its dose; the inclusion and exclusion criteria of the study; the concomitant immunosuppression; the outcomes and the duration of the study were reviewed. The two most common drugs used for the treatment of SSc-ILD are cyclophosphamide and mycophenolate mofetil, both supported by randomized controlled trials. Additional drugs, such as nintedanib and tocilizumab, have been approved to slow pulmonary function decline in SSc-ILD. In this review, we discuss the therapeutic alternatives for SSc management, offering the option to customize the design of future studies to stratify SSc patients and provide a patient-specific treatment according to the new emerging pathogenic features of SSc-ILD.

## 1. Introduction

Systemic sclerosis (SSc) is a complex rare autoimmune disease with heterogeneous clinical manifestations, including vasculopathy, immune dysfunction, musculoskeletal inflammation, and fibrosis of the skin and internal organs [1,2]. Currently, interstitial lung disease (ILD) and cardiac involvement (including pulmonary arterial hypertension) are recognized as the leading causes of SSc-associated mortality; in particular, ILD appears to be virtually always present in SSc patients according to autopsy studies, with up to 90% having evidence of pulmonary involvement on HRCT, with 40–75% showing reduced pulmonary function tests [3]. Indeed, interstitial lung disease accounts for up to a third of mortality causes [4,5], thus representing the leading cause of death in SSc. The discovery of targeted therapies is still an unmet clinical need, due to the complex multifactorial pathogenesis.

SSc-ILD progression is commonly characterized by a slow rate of progression, with about a quarter of SSc patients experiencing ILD progression in 1 year time and around a third in 5-year time [6]. A number of candidate therapies are under evaluation and, simultaneously, ongoing trials are also numerous, including phase I and II studies. 

Cyclophosphamide (CYC) and mycophenolate mofetil (MMF) are the two most common drugs used for the treatment of SSc-ILD; their use is supported by randomized controlled trials (RCT) [7,8] that demonstrated similar effective results, although MMF has less risk to fertility, favorable ease of follow-up, with a reduced risk of secondary malignancies. Nevertheless, the latest SSc-ILD treatment guidelines recommend CYC and hematopoietic stem cell transplant considering these therapeutic approaches are supported by completed RCTs [9]. Additional drugs, such as nintedanib and tocilizumab, have been approved to slow down pulmonary function decline in SSc-ILD [10,11,12].

In this context, the aim of the present literature review is to analyze the phase II and phase III SSc-ILD randomized clinical trials published in the last 5 years. 

Full peer-reviewed manuscripts reporting phase II, phase III, and head-to-head randomized clinical trials regarding SSc-ILD studies, published from 1 January 2016 to 31 December 2021 and describing outcomes following pharmaceutical-based interventions, were included. Studies with heterogeneous ILD populations (e.g., connective tissue disease-associated ILD (CTD-ILD)) as well as study designs, case reports, review articles, letters to the editor, editorials, preclinical studies, non-pharmacological interventions, and guidelines were excluded from the analysis. 

Study design with sample size, treatment regimen details, participant baseline characteristics, study endpoints, assessment of pulmonary function, patient function or quality of life measures, survival and safety outcomes were reported in a descriptive table for phase II, phase III, and head-to-head randomized clinical trials. 

## 2. Randomized Clinical Trials

### 2.1. Phase III

#### 2.1.1. Nintedanib 

Nintedanib, a multi-target tyrosine kinase inhibitor, is a small molecule designed as an ATP-competitive inhibitor of platelet-derived growth factor receptor (PDFGR), fibroblast growth factor receptor (FGFR), and vascular endothelial growth factor receptor (VEGFR), recently approved for the treatment of idiopathic pulmonary fibrosis (IPF) [13]. SSc-ILD and IPF share some similar fibrogenic mechanisms that include fibroblast activation, migration, proliferation, and differentiation into myofibroblasts, resulting in excessive collagen deposition [14]. The phase III, multi-center, randomized, double-blind, placebo-controlled SENSCIS (Safety and Efficacy of Nintedanib in Systemic SClerosIS) trial evaluated nintedanib safety and efficacy in patients affected by SSc-ILD (NCT02597933).

The primary endpoint was the annual rate of forced vital capacity (FVC) decline (ml/year) assessed over 52 weeks [15]. In the primary end-point analysis, the adjusted annual rate FVC change was −52.4 mL/year in the treated group and −93.3 mL/year in the placebo group (difference, 41.0 mL per year; 95% confidence interval (CI), 2.9 to 79.0; *p* = 0.04). The adjusted mean annual rate of FVC change as a percentage of the predicted value at week 52 was −1.4% in the nintedanib arm and −2.6% in the placebo arm (difference, 1.2 percentage points: 95% CI, 0.1 to 2.2). Patients in the arm treated with MMF at baseline had a better performance. Furthermore, the decrease in FVC differed in the placebo group and it is associated with MMF use. Although a randomization according to MMF administration was not used, these data suggest a potential additional benefit of MMF use with nintedanib on lung function [10]. However, additional studies are necessary to explore the combined use of MMF and nintedanib in SSc-ILD patients. In a post hoc analysis of the SENSCIS trial, the authors assessed the proportions of patients with categorical changes in % predicted FVC at week 52 and the time to absolute decline in FVC of ≥5% predicted or death and absolute decline in FVC of ≥10% predicted or death. Over 52 weeks, the hazard ratio (HR) for an absolute decline in FVC of ≥5% predicted or death with nintedanib versus placebo was 0.83 (95% confidence interval [95% CI] 0.66−1.06) (*p* = 0.14), and the HR for an absolute decline in FVC of ≥10% predicted was 0.64 (95% CI 0.43−0.95) (*p* = 0.029), confirming that nintedanib has a clinically significant advantage in reducing the progression of SSc-ILD [16].

#### 2.1.2. Tocilizumab

Tocilizumab (TCZ) is a recombinant humanized monoclonal antibody of the immunoglobulin G1k subclass, that binds interleukin 6 (IL-6) receptor, thus blocking its signaling. The molecule is genetically engineered and is produced by grafting the complementarity determining region of mouse anti-human IL-6 R to human IgG1 [17]. TCZ is approved in Europe for the treatment of selected forms of rheumatoid arthritis and analogous diseases non-responsive to methotrexate or other disease-modifying anti-rheumatic drugs and it is under evaluation in other diseases sharing inflammatory pathogenesis [18]. The rationale for targeting IL-6 starts from the observation of high levels of IL-6 in skin and lung tissues of SSc-patients [19,20]; moreover, IL-6 levels tightly correlate with skin thickness scores supporting a causal relationship [21]. Starting from these data the faSScinate trial (NCT01532869) was designed. FaSScinate is a phase 2, randomized, double-blind, placebo-controlled trial enrolling 87 SSc patients [11]. Patients were assigned into different groups using randomization (1:1) to receive weekly subcutaneous treatment with TCZ 162 mg or placebo for 48 weeks followed by open-label weekly TCZ for additional 48 weeks. The primary efficacy endpoint was the difference in mean change from baseline in modified Rodnan skin score (mRSS) at week 24. Despite the primary endpoint was not achieved, with a difference of −2.70 mRSS units (95% CI: −5.85, 0.45) in favor of TCZ and in absence of a statistical significance (*p* = 0.0915), the study evidenced that fewer patients in the TCZ arm had a decline in % predicted FVC than in the placebo arm with respect to cumulative distribution (week 48, *p* = 0.0373). Overall, after 48 weeks of treatment, safety in faSScinate was coherent with the natural history of SSc and the known safety profile for TCZ [10]. Following these encouraging results, a phase 3 study, the focuSSced trial (NCT02453256), was designed [22]. The primary endpoint was the difference in change from baseline to week 48 in modified Rodnan skin score (mRSS), while % predicted FVC, time to treatment failure, and patient-reported and physician-reported outcomes were secondary endpoints. The change in mRSS was higher in the TCZ arm (−6.1 vs. −4.4) but it did not meet significance (*p* = 0.1). On the other hand, the TCZ group showed a significant stabilization of FVC compared to placebo, according to the subgroup analysis that considered only SSc participants with ILD (−14 mL vs. −255 mL at 48 weeks, *p* < 0.001). These findings were also associated with a reduced number of SSc-ILD patients having a ≥10% decline in % predicted FVC in the TCZ compared to placebo (5 vs. 14). Taken all together, these results support the use of TCZ in SSc patients with early ILD, as already approved by the Food and Drug Administration. 

#### 2.1.3. Lenabasum

Lenabasum is a synthetic cannabinoid receptor type-2 (CB2) agonist recognized as an inflammation-resolving drug candidate for the treatment of different diseases, thanks to its anti-inflammatory effects, and also for systemic sclerosis [23]. A randomized, double-blind, placebo-controlled, phase II study (NCT02465437) was designed to evaluate the safety and tolerability of lenabasum in patients with active SSc [24]. Lenabasum treatment was safe, well-tolerated, and improved multiple efficacy assessments of overall disease, such as skin involvement and patient-reported outcomes. FVC was used as a lung performance measure and was associated with numerical improvement from baseline in the lenabasum group compared to the placebo group starting at week 8, with a maximal but non-significant mean ± SEM treatment difference of 1.7 ± 1.6% observed at week 12. These data encouraged authors to provide other analyses and patients who had completed the 16-week phase 2 study were enrolled to continue lenabasum treatment at the dose of 20 mg twice a day. Thirty-six patients were enrolled and 26 patients were treated for >92 weeks. Predicted FVC values declined by 3.2% from the start of the study, but the trial design of this “open period” limits the conclusion. Despite the interesting results, the study shows some limitations, including the small number of patients, short-term observation period, and the subsequent open phase. RESOLVE-1 (NCT03398837) was the phase 3 study [25]. This study presents some novel aspects with respect to other studies: the American College of Rheumatology (ACR) Combined Response Index in diffuse cutaneous Systemic Sclerosis (CRISS) score was assigned as the primary outcome and interim analysis has been presented at the 2021 American College of Rheumatology meeting [26]. The CRISS score involves a two-step process: the first step is to identify relevant disease worsening or the occurrence of new-onset end-organ damage. The second step requires calculating the probability of patient improvement after 1 year of therapy based on a scale ranging from 0 to 1 point according to changes from baseline status in five domains: the mRSS, % predicted FVC, patient and physician global assessments, and the Health Assessment Questionnaire Disability Index (HAQ-DI). A responder is defined as a patient having a CRISS score of 0.6 or higher and no significant renal or cardiopulmonary worsening [27]. Despite in the RESOLVE-1 trial the primary outcome was not achieved, post hoc analysis suggests that patients taking lenabasum in association with background MMF for more than 2 years were more likely to have stable % predicted FVC over 1 year compared to MMF and placebo (64% vs. 35%). These findings, however, need to be confirmed in future studies.

### 2.2. Phase II

#### 2.2.1. Pirfenidone

Pirfenidone is a small molecule comprising a modified phenyl pyridine with pleiotropic anti-fibrotic, antioxidant and anti-inflammatory properties, although its exact mechanism of action remains unclear. Pirfenidone was initially identified as an anti-inflammatory compound in animal models [28]. However, the unexpected identification of anti-fibrotic effects in animals treated with pirfenidone redefined the interest in this molecule [29]. Pirfenidone has been shown to attenuate fibrosis in different organs, including lung, liver, heart, and kidney [30]; it is actually approved for IPF treatment [31] and has also been tested in SSc-ILD patients in the LOTUSS study (an Open Label, RandOmized, Phase 2 STUdy of the Safety and Tolerability of Pirfenidone when Administered to Patients with Systemic Sclerosis-Related Interstitial Lung Disease) (NCT01933334) [32]. However, pirfenidone use is known to be associated with adverse events (AE) related to the liver, skin, and gastrointestinal (GI) system which are also frequently observed in SSc patients. Reasonably, the primary endpoint was the safety of this treatment in SSc-ILD patients, whereas the secondary endpoint was the effect of pirfenidone on predicted FVC and diffusion capacity of carbon monoxide (DLCO). Data showed an acceptable tolerability profile of pirfenidone in SSc-ILD and this tolerability was not affected by concomitant MMF use. However, FVC and DLCO values were basically unchanged throughout the observation period and no clinically relevant differences were observed in lung function variables between the groups or in any of the subgroup analyses. Lately, another double-blind, randomized, placebo-controlled, pilot study was conducted to evaluate the safety and efficacy of pirfenidone in SSc-ILD. Analogous to the LOTUSS trial, this study was unsuccessful in finding a relevant beneficial effect of pirfenidone over placebo in improving/stabilizing FVC, exercise capacity, symptoms, or skin disease. Nevertheless, the trial is underpowered to allow conclusive evidence [33]. 

#### 2.2.2. Pomalidomide

Pomalidomide (POM) is an immunomodulatory agent structurally similar to thalidomide and lenalidomide. POM had shown immune-modulating activity on myeloid and lymphocyte cells and exhibited anti-fibrotic effects in pre-clinical models of dermal fibrosis [34]. Furthermore, it enhances T cell and natural killer (NK) cell-mediated immunity and inhibits pro-inflammatory cytokines release, such as tumor necrosis factor (TNF) or IL-6. In a study including 11 SSc patients, histologic comparison of skin biopsies showed changes in skin fibrosis and an increase in epidermal and dermal infiltrating CD8 (+) T cells following thalidomide treatment. Moreover, thalidomide reduced IL-12 and TNF plasma levels. These changes were associated with clinical effects, including dry skin, dermal edema, transient rashes, and healing of digital ulcers [35]. Starting from these data and from a similar structure, POM was tested in SSc patients. Due to difficulties in recruiting patients for the study owing to restrictive inclusion and exclusion criteria, the sponsor terminated enrollment. According to the interim analysis data, the study did not show a significant amelioration in any of the co-primary efficacy endpoints (changes from baseline in FVC and mRSS) for patients who concluded blinded treatment [36]. 

#### 2.2.3. Romilkimab

Romilkimab is a bispecific monoclonal (immunoglobulin-G4) antibody that binds and neutralizes both IL-4 and IL-13 [37]. IL-4 and IL-13 are Th2-derived cytokines involved in the pro-fibrotic mechanisms of SSc; in fact, their increased levels have been detected both in serum and in skin biopsies of SSc patients [38]. A phase 2A, randomized, double-blind, placebo-controlled, 24-week trial was performed in SSc patients (NCT02921971) [39] that focused the primary endpoint on evaluating the change from baseline to week 24 in modified mRSS and the secondary endpoints on FVC and DLCO. Romilkimab resulted in a significant reduction in mRSS from baseline to week 24 versus placebo (−4.32 to −0.31; *p* = 0.0291. The least square mean (SE) change in FVC was −10 (40) mL for romilkimab versus −80 (40) mL for placebo at week 24 resulting in a non-significant mean (SE) difference (95% CI) of 70 (60) mL (−40 to 190; *p* = 0.10). In this study, romilkimab had a non-significant but favorable effect on lung outcomes, which might justify further assessment. Moreover, in patients treated with a placebo, the evidence of the loss of 80 mL for FVC between baseline and week 24 might supports the hypothesis that patients with early SSc may develop significant lung disease in a very short period of time. 

#### 2.2.4. Riociguat

The soluble guanylate cyclase (sGC) stimulator riociguat increases intracellular cyclic guanosine monophosphate (cGMP) and consequently activates protein kinases G, with an effect on the regulation of vascular tone and remodeling [40]. Riociguat was approved for the treatment of pulmonary arterial hypertension [41]. A 52 week, double-blind, placebo-controlled, multi-center, randomized phase 2 study (RISE-SSc, NCT0228376219) was performed in recently diagnosed SSc patients (disease duration ≤ 18 months) in order to investigate the potential effects of riociguat. The primary endpoint was the change in mRSS from baseline to week 52 whereas among secondary endpoints, change in FVC% was evaluated. Among secondary endpoints, DLCO% and FVC% were evaluated overall and (post hoc) in patients with ILD according to medical history and restrictive lung disease (FVC% 50–75% at baseline). The modification in FVC% between baseline and week 52 was −2.38% (SD 7.52) with riociguat and −2.95% (SD 9.73) with placebo (difference of LS means −0.20 (SE 1.61); 95% CI −3.40 to 3.00; nominal *p* = 0.901). Depending on the diagnosis (patients with ILD or patients with restrictive disease) the mean change in FVC% from baseline to week 52 was −7.6 to −8.7% with placebo and +0.7 to −2.7% with riociguat. Some measures of mRSS, lung function in patients with evidence for pre-existing ILD and the prevention of new Raynaud’s phenomenon and digital ulcers symptoms suggest potential signals for efficacy. It is important to highlight that the results of descriptive analyses of secondary and exploratory endpoints should not be interpreted as the efficacy of riociguat, but as a potential indicator to be examined in further studies [42]. 

#### 2.2.5. Rituximab 

Rituximab (RTX) is a chimeric murine/human monoclonal antibody targeted against CD20, a specific pan-B-cell marker and the only binding site for RTX. RTX binding to the cell surface results in the destruction of lymphocytes through several mechanisms including apoptosis activation, complement-dependent cytotoxicity, or antibody-dependent cytotoxicity.

B cells have been found in the skin and lungs of patients with SSc-ILD, and an increased expression of B cell-related genes has also been demonstrated in the skin [11]. Moreover, in SSc patients, the B cell population is predominantly represented by naïve B cells compared to memory B cells, although the latter are highly active [43].

A multi-center, open-label, comparative study was performed on a total of 51 patients with SSc-ILD and treated with rituximab (RTX) or conventional treatment (azathioprine, methotrexate, and MMF) [44]. Patients treated with RTX showed an increase in FVC at 2 years (mean ± SD of FVC: 80.60 ± 21.21 vs. 86.90 ± 20.56 at baseline vs. 2 years, respectively, *p* = 0.041 compared to baseline). Patients in the control group had no significant change in FVC during the first 2 years of follow-up. At the 7th year, the remaining patients in the RTX group had higher FVC compared to baseline (mean ± SD of FVC: 91.60 ± 14.81, *p* = 0.158 compared to baseline) in contrast to patients in the control group that showed an FVC deterioration (*p* < 0.01, compared to baseline). Direct comparison between the 2 groups showed a significant benefit for the RTX group in FVC (*p* = 0.013).

Of note, a recent trial analyzed the efficacy and safety of RTX in SSc patients. This was a phase II, multi-center, double-blind, parallel-group, investigator-initiated, randomized, placebo-controlled trial (DESIRES) conducted in four Japanese hospitals. The results, despite the limited number of patients, are relevant demonstrating that RTX at the dose of 375 mg/m^2^ every week for 4 weeks produces a significant improvement in mRSS compared to placebo. Additionally, among the secondary endpoints included, FVC% appears to be stable in RTX-treated overall compared to the reported reduction of the placebo, with promising results reported in the subgroup analysis regarding SSc-ILD patients [45]. 

#### 2.2.6. Abatacept 

Based on growing evidence suggesting a key role for T cells in the development of both skin and internal organ damage in systemic sclerosis [46,47], a phase II investigator-initiated, multi-center, double-blind randomized placebo-controlled trial was designed to assess the safety and efficacy of abatacept in 88 early diffuse SSc [48]. mRSS change at 12 months, which was the primary endpoint of the study, was not statistically significant; neither was % predicted FVC. Nonetheless, a significant difference was observed in HAQ-DI and ACR CRISS between abatacept and placebo. These results, together with the superior safety profile of abatacept compared to placebo and the higher number of SSc patients requiring rescue therapy in the placebo group, led to the extension of this trial with an additional 6-month open-label, double-blind, randomized trial [49]. Overall, the authors concluded that both mRSS and FVC% predicted showed numerical improvement in patients assigned abatacept compared with those assigned to placebo which was not statistically significant; moreover, substantial individual heterogeneity should be considered. 

### 2.3. Head-to-Head

#### 2.3.1. Rituximab vs. Cyclophosphamide

RTX has been used in patients with SSc with pulmonary and renal involvement and has shown efficacy in patients refractory to CYC [44,50,51]. A prospective, randomized, open-label, parallel-group trial was performed including SSc patients with skin and lung involvement [52]. This trial offers the unique feature to provide a head-to-head study, including CYC as a comparator for the assessment of endpoints. The primary outcome was the % predicted FVC value at 6 months. There was a significant improvement in the predicted FVC in the RTX group [from 61.30 (11.28) at baseline to 67.52 (13.59) at the end of the study; *p* = 0.002]. The mean difference in predicted FVC was 9.46 (95% CI: 3.01, 15.90; *p* = 0.003) in favor of the RTX group. This study met its primary endpoint, demonstrating that RTX improved the % predicted FVC, while CYC was not associated with a % precited FVC stability after 6 months. In fact, % predicted FVC in the RTX group significantly increased from 61.30 at baseline to 67.52 at the end of the study, while in the CYC group FVC showed a non-significant decrease from 59.25 at baseline to 58.06. In conclusion, the authors described a greater increase in % predicted FVC value as well as decreased mRSS in the RTX arm compared with the CYC arm. Since the mean difference in % predicted FVC was in favor of the RTX arm (9.46; 95% CI: 3.01, 15.90; *p* = 0.003) and the lower limit of 95% CI of the mean difference of % predicted FVC was 3.01, this study met the non-inferiority criterion vs. CYC (margin 2%). Furthermore, RTX therapy was associated with a good safety profile. 

#### 2.3.2. Mycophenolate Mofetil vs. Cyclophosphamide

Because of its immunosuppressive activity and an acceptable safety and tolerability profile, MMF has been widely employed in uncontrolled studies for the treatment of SSc-ILD in the last years. In fact, Tashkin et al. designed a multi-center randomized, double-blind, clinical trial in order to assess the efficacy and safety of MMF using CYC as a comparator drug in the Scleroderma Lung Study II (SLS II) [8], as the natural evolution of the SLS I [7]. In this trial, 69 SSc-ILD symptomatic patients were treated with MMF for 24 months and 73 with CYC for 12 months followed by placebo for additional 12 months. This preliminary hypothesis was based on the results of SLS I, thus assuming that MMF would be effective as CYC at 18 months, while the 12 months treatment with CYC would be associated with a fall in %-predicted FVC back to untreated values after additional 12 months of placebo treatment. While this trial failed to demonstrate MMF superiority, the results showed that MMF is also not inferior to CYC in producing an improvement in lung function, though modest and measured in an increase in % predicted FVC of 3.0 ± 1.2% and 3.3 ± 1.1% within the CYC and MMF treatment arms, respectively. Additionally, MMF induced significant between-treatment differences in DLCO %-predicted and DL/VA %-predicted supporting a slower decline during the MMF treatment than CYC. 

Regarding safety, in the CYC arm, hematopoietic suppression occurred more frequently. On the other hand, the number of pneumonias, infections, systemic adverse events, or deaths, was not different between the treatment arms. Nevertheless, a greater percentage of subjects in the CYC arm prematurely discontinued the drug and the tolerated dose of CYC decreased over time to approximately 75% of the target dose.

## 3. Discussion

SSc exhibits a complex heterogeneity in multisystem organ involvement due to multiple cellular and molecular interactions. A growing body of data identified several potential targets, leading to a better definition of the pathogenesis of the disease. In the last 5 years, SSc-ILD, as the leading cause of mortality among SSc patients, attracted numerous interventional trials with a design improved for stratification, screening, timing, and evaluation [53,54]. 

In the present critical review of published phase II, phase III, and head-to-head clinical trials from the last 5 years, we found high-quality trials regarding mycophenolate mofetil, rituximab, riociguat, romilkimab, pomalidomide, pirfenidone, abatacept, lenabasum, nintedanib, and tocilizumab (Table 1, Table 2 and Table 3). These molecules act at different levels in the fibrotic and immunologic pathways of SSc-ILD (Figure 1). 

From the comparison of these clinical trials, it appears evident that most of the results related to SSc-ILD and pulmonary function are extrapolated from clinical trials where the primary endpoint is the progression of skin disease, thus patients’ selection criteria are not defined specifically for SSc-ILD [55]. A very elegant score, attempting to identify a core set of variables to include in each clinical trial in SSc, is the ACR CRISS and its revised version for response (rCRISS), which has been used recently and encompasses, apart from mRSS and FVC, also patient-and physician-reported outcomes in a single score [56].

As evidenced in Table 2, which shows the trials in the most advanced phase, the primary endpoint is always different and it does not allow to perform an ideal comparison. The nintedanib trial offers the ideal patients’ selection, restricted to SSc-ILD, without needing to provide sub-group analysis, such as for tocilizumab and lenabasum. 

In addition, the enrollment of patients exposed to background immunosuppressive therapy, allowed in the nintedanib and lenabasum trials, and the addition of rescue therapy during the tocilizumab trials at week 16 if needed, further complicates the analysis of the results. 

Ideally, the trial design of a study regarding SSc-ILD should provide the additional or cumulative effect of a drug over MMF, based on the results of previous trials confirming its non-inferior efficacy to CYC, already modest [7], and its superiority to CYC in terms of tolerability and treatment discontinuation [8]. Indeed, after the results obtained by MMF in the phase II trial, and the promising data from pirfenidone trial in SSc [33], a randomized clinical trial, namely the Scleroderma Lung Study III, has been designed (NCT03221257), aiming at assessing the efficacy and safety of the combination of the anti-fibrotic effects of pirfenidone with mycophenolate for the treatment of SSc-ILD. 

However, it remains difficult to identify and stratify SSc-ILD patients according to the most appropriate target or combination therapy, as supporting evidence to employ anti-fibrotic vs. immunosuppressant medications is lacking. An elegant attempt to suggest a patient stratification strategy, based on autoantibodies profile and skin gene profiling could prove useful in driving the most appropriate therapeutic combination [57]. 

Rituximab could be another candidate for future trials in SSc-ILD employing a combination therapy design, as this drug showed a better safety profile in a head-to-head study vs. CYC and the stabilization of pulmonary function in a randomized clinical trial compared to placebo [58]. In fact, recent reports support RTX use in association with MMF [59,60]. Similarly, a subgroup analysis of the nintedanib trial has been published reporting the post hoc analysis performed to estimate the proportion of patients with an absolute decrease in FVC of at least 3.3% predicted at week 52, considered as the minimal clinically important difference estimate for worsening of FVC in patients with SSc-ILD, by concomitant MMF use at baseline [61]. This study suggests that combination therapy might have additional beneficial effects for the treatment of SSc-ILD. 

This emerging evidence suggests that the default consensus reference drug for SSc-ILD is MMF [10,62,63], as supported by the recent post hoc analysis of the nintedanib trial [61].

Since the sequencing of immunosuppressants versus anti-fibrotic treatments remains to be tested, further combination studies are needed, as already proposed for idiopathic pulmonary fibrosis [31], to clarify if the combination is superior to either one alone. 

Revised ACR CRISS (rCRISS) responses: the proportion of participants that improve in ≥3/5 core items by certain percentages (30%, except ≥5% in FVC%).

## 4. Conclusions

The primary outcome in ILD studies is to halt disease progression and avoid irreversible lung damage and pulmonary function deterioration. Among the most recent phase III trials, nintedanib and tocilizumab achieved this goal with different endpoints, while the results of the lenabasum study are not available [22,61]. Furthermore, the promising results reported for rituximab, according to the recent phase II trial and the results of the study in comparison with CYC, warrant a confirmatory long-term phase III trial for extensive use. While several published [10,16,62] and upcoming (NCT02370693-NCT03221257) trials confirm the use of MMF as the primary drug in SSc-ILD combination therapy, several additional cellular and molecular targets have been investigated in placebo-controlled trials. 

On one hand, the trials regarding pomalidomide, an analog of thalidomide, and abituzumab, a pan-αν integrin inhibiting monoclonal antibody, were interrupted for recruiting issues. On the other hand, subgroup analyses show promising effects on pulmonary function for romilkimab, an anti-interleukin-4/interleukin-13 monoclonal antibody, abatacept, an antibody against costimulatory molecules CD80 and CD86, and riociguat, a cGMP inhibitor. Taken together, these phase II trials confirm that an essential unmet need in SSc-ILD studies is a better definition of the eligible population and a definitive identification of treatment arms for combination therapy and duration of treatment exposure. 

## 5. Future Directions

While there is a general consensus based upon recommendations on the use of immunosuppressants in early SSc to treat cutaneous progression [9], the choice of the most appropriate drug for SSc-ILD between antifibrotic and immunosuppressants remains challenging.

Hopefully, ongoing or upcoming trials, such as the combination of bortezomib and MMF (NCT02370693) or the combination of pirfenidone and MMF (Scleroderma Lung Study III—NCT03221257), with a trial design dedicated to SSc-ILD, will provide new alternatives, increasing the quality of evidence already available, with a precise definition of patients’ selection criteria, background immunosuppressive therapy, and treatment duration.

We now have growing therapeutic alternatives for SSc management, offering the option to customize the design of future studies to stratify SSc patients and provide a patient-specific treatment according to the new emerging pathogenic features of SSc-ILD.

Further research is required to identify the therapeutic algorithm to support combination therapy, improve criteria for patient enrollment in clinical trials, and provide the optimal timing for treatment initiation to achieve the ambitious endpoint of improving and stabilizing over time pulmonary function.

## Figures and Tables

**Figure 1 biomedicines-10-00504-f001:**
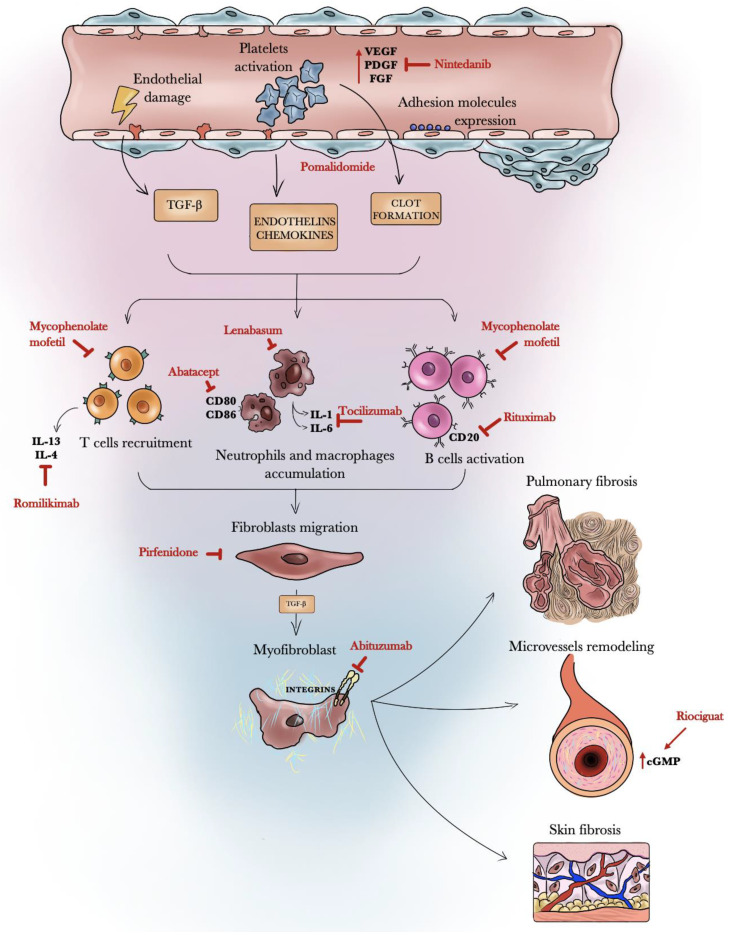
Schematic representation of the mechanism of action of nintedanib, pomalidomide, tocilizumab, rituximab, romilikimab, pirfenidone, lenabasum, and riociguat.

**Table 1 biomedicines-10-00504-t001:** Phase II randomized double-blind clinical trials in SSc-ILD.

Molecule and Trial	Study Design	Endpoints	Efficacy	Safety
Tocilizumab (FaSScinate) (NCT01532869)	Randomization 1:1 TCZ (*n* = 43) 162 mg (s.c. weekly) Placebo (*n* = 44) Duration 24 weeks followed by open-label weekly TCZ for additional 24 weeks.	Primary: mRSS Secondary: patient-reported and physician-reported outcomes % predicted FVC, % predicted DLCO	The primary endpoint not reached (treatment difference of −2.70 mRSS units (95% CI: −5.85, 0.45) in favour of TCZ, with no statistical significance (*p* = 0.0915). Fewer patients in the TCZ arm had a decline in FVC% than in the placebo arm (week 48, *p* = 0.0373).	42/43 (98%) patients in the TCZ group vs. 40/44 (91%) in the placebo group had adverse events. 14 (33%) vs. 15 (34%) had serious adverse events. Serious infections were more common in the TCZ group (7/43 (16%)) than in the placebo group (2/44 (5%)). One patient died in the TCZ group
Lenabasum (NCT02465437)	Randomization: 2:1 Lenabasum (*n* = 27) (5 mg once daily, 20 mg once daily, or 20 mg twice daily for 4 weeks, followed by 20 mg twice daily for 8 weeks) Placebo (*n* = 15) Duration: 12 weeks	Primary: CRISS score	The median CRISS score increased in the lenabasum group during the study, reaching 0.33, versus 0.00 in the placebo group, at week 16 (*p* = 0.07 by 2-sided mixed-effects model repeated-measures analysis).	9/15 (60%) of the placebo-treated subjects and 17/27 (63%) of the lenabasum-treated subjects had adverse events (AEs). No serious or severe AEs related to lenabasum were observed. No deaths.
Pirfenidone (LOTUSS) (NCT01933334)	Randomization: 1:1 Pirfenidone starting dose: 801 mg/day-maintenance dose of 2403 mg/day 2 week titration (*n* = 32) 4 week titration (*n* = 31) Duration: 16 weeks	Safety	Data showed an acceptable tolerability profile of pirfenidone in SSc-ILD, and tolerability was not affected by concomitant MMF use. FVC% and DLCO values were unchanged throughout the observation period, and no clinically relevant differences were observed in lung function variables between the groups or in any of the subgroup analyses	96.8% experienced an adverse event 6 patients discontinued early for treatment-related adverse events in the pirfenidone group.
Pirfenidone (NCT03856853)	Randomization: 1:1 Pirfenidone (*n* = 17) 2400 mg/day Placebo (*n* = 17) Duration: 24 weeks	Primary: stabilisation or improvement in FVC Secondary: absolute change in the % predicted FVC, Mahler’s dyspnoea index, 6MWD, MRSS and TNF and TGF-β serum levels	Stabilisation/improvement in FVC was seen in 94.1% and 76.5% subjects in the pirfenidone and placebo groups, respectively (*p* = 0.33). The median (range) absolute change in % predicted FVC was −0.55 (−9 to 7%) and 1.0 (−42 to 11.5%) in the treatment and control groups, respectively (*p* = 0.51). The changes in 6MWD, dyspnoea scores, MRSS, and levels of TNF and TGF-β were not significantly different between groups.	Adverse events were common among the groups (96.1% in the prifenidone and 100% in the placebo group) Gastrointestinal intolerance led to discontinuation of the drug in two patients in the pirfenidone group and one of these subjects required hospitalisation
Pomalidomide (NCT01559129)	Randmization 1:1 POM (*n* = 11) 1 mg QD Placebo (*n* = 12) Duration: 52 weeks (discontinued early)	Primary: mRSS, % predicted FVC and UCLA Scleroderma Clinical Trial Consortium Gastrointestinal Tract	Because of recruitment challenges, subject enrollment was discontinued early. Interim analysis showed that primary endpoints were not met	POM was generally well tolerated, with an adverse event profile consistent with previous studies regaridng POM
Romilkimab (NCT02921971)	Randmization 1:1 Romilkimab (*n* = 48) 200 mg s.c. weekly Placebo (*n* = 49) Duration: 24 weeks	Primary: mRSS. Secondary: HAQ-DI, observed FVC/observed DLCO.	Romilkimab resulted in a statistically significant decrease in mRSS versus placebo. FVC and DLCO show a positive trend for romilkimab without reaching statistical significance	Overall incidence of treatment-emergent AEs (TEAEs) was high (>80% in both groups. Mild or moderate in intensity (40%) and severe (2%) for romilkimab; and 76% mild or moderate and severe (8%) for placebo
Riociguat (RISE-SSc) (NCT0228376219)	Randomization 1:1 Riociguat (*n* = 60) individually adjusted every 2 weeks from 0.5 mg to 2.5 mg orally three times daily over 10 weeks Placebo (*n* = 61) Duration: 52 weeks	Primary: mRSS Secondary: ACR CRISS, HAQ-DI, patient’s global assessment, physician’s global assessment and change in FVC%.	Riociguat did not show significantly benefit in mRSS versus placebo at the predefined *p* < 0.05. Overall, the change in FVC% was not significant. Subgroup analysis suggests potential signals for efficacy	Overall, 96.7% patients in the riociguat group and 55 90.2% in the placebo group experienced an AE. Severe adverse events were less common with riociguat than with placebo
Abituzumab (STRATUS) NCT02745145)	Randomization: 2:2:1 24 SSc-ILD patients on stable mycophenolate Abituzumab 1500 mg, abituzumab 500 mg, or placebo every 4 weeks Duration: 104 weeks	Annual rate of change in absolute forced vital capacity	terminated prematurely	Well tolerated No new safety signals were detected.
Abatacept (ASSET) (NCT02161406)	Randomization: 1:1 Abatacept (*n* = 44) 125 mg s.c. weekly Placebo (*n* = 44) no background immunomodulatory therapies were allowed apart steroids	Primary: mRSS Secondary: 28-swollen and tender joint count, patient global assessment for overall disease, physician global assessment for overall disease, PROMIS-29 v2 Profile, HAQ-DI, Scleroderma-HAQ-DI VAS pain, burden of digital ulcers and Raynaud’s, UCLA GIT 2.0; FVC% predicted, ACR-CRISS	The primary outcome measure was not statistically significant. According to intrinsic gene expression subset based on a machine learning classifier. The fibroproliferative subset showed a numerical increase in FVC% in the abatacept arm (*p* = 0.19) while all other groups showed decreases in FVC%.	Abatacept was found to be generally safe with no new safety signals, with lower numbers of participants experiencing AEs, infectious AEs, and SAEs compared to the placebo group
Rituximab (DESIRES) (NCT04274257)	Randomization 1:1 Rituximab (*n* = 28) 375 mg/m² once a week for 4 weeks Placebo (*n* = 28) Duration: 24 weeks	Primary: mRSS Secondary: % predicted FVC, predicted DLCO%, TLC, SF-36, HAQ-DI.	The absolute change in mRSS was significantly lower in the rituximab group than in the placebo group (−6.30 vs. 2.14; difference −8.44 (95% CI −11.00 to −5.88); *p* < 0.0001). Absolute change in % predicted FVC was statistical signficant (0.09% in the rituximab group vs. 2.87% in the placebo group)	Adverse events were almost similar in both groups (100% in the rituximab and 88% in the placebo group) One serious adverse event leading to treatment discontinuation occurred in one patient in each group. Upper respiratory infections occurred in 39% rituximab-treated patients and in 38% of the placebo-treated patients There were no deaths during follow-up.

CRISS: American College of Rheumatology Combined Response Index in diffuse cutaneous Systemic Sclerosis (CRISS) score: MRSS, Health Assessment Questionnaire (HAQ) disability index (DI), physician global assessment of overall patient health, patient global assessment of health, FVC%.

**Table 2 biomedicines-10-00504-t002:** Phase III randomized double-blind clinical trials in SSc-ILD.

Molecule and Trial	Study Design	Endpoints	Efficacy	Safety
Tocilizumab (FocuSSced) (NCT02453256)	Ranzomization 1:1 with IL-6 stratification TCZ (*n* = 104) 162 mg (s.c. weekly) Placebo (*n* = 106) Duration: 48 weeks	Primary: mean change difference in mRSS. Secondary: % predicted FVC, time to treatment failure, patient-reported and physician-reported outcomes	The change in mRSS was higher in TCZ arm (−6.1 vs. −4.4) but not significant (*p* = 0.1). TCZ group showed instead a significant slower reduction of FVC% compared to placebo, and the results appear more significant in the subset with SSc-ILD with a stabilization of FVC% contrasting to the decline in the placebo group (−20 mL vs. −257 mL, *p* < 0.001)	Infections were the most common adverse events (52% in the TCZ group vs. 50% in the placebo group). Serious adverse events were reported in 13 participants treated with TCZ and 18 with placebo, primarily infections and cardiac events.
Nintedanib (SENSCIS) (NCT02597933)	Randomization: 1:1 with anti-topoisomerase stratification Nintedanib (*n* = 288) 150 mg (orally twice daily) Placebo (*n* = 288), SSc-ILD patients receiving prednisone (≤10 mg/day) and/or a stable dose of MMF or methotrexate, were considered eligible. Duration: 52 weeks	Primary: annual rate of decline in FVC (ml/year) Secondary: absolute changes from baseline in the modified Rodnan skin score and the total score on the St. George’s Respiratory Questionnaire (SGRQ)	The adjusted annual rate of change in FVC was −52.4 mL/year in the treated group and −93.3 mL/year in the placebo group (difference, 41.0 mL per year; 95% confidence interval (CI), 2.9 to 79.0; *p* = 0.04). The adjusted mean annual rate of change in % precited FVC at week 52 was −1.4% in the nintedanib arm and −2.6% in the placebo arm (difference, 1.2 percentage points; 95% CI, 0.1 to 2.2).	Discontinuation was higher in the nintedanib group than in the placebo group (16% vs. 8.7%) Diarrhea, the most common adverse event, was reported in 75.7% of the patients in the nintedanib group and in 31.6% of those in the placebo group. 3.5% in the nintedanib group and 3.1% in the placebo group died
Lenabasum (RESOLVE-1) (NCT03398837) Data from interim analysis	Randomization 1:1:1 Lenabasum (*n* = 100) 20 mg orally twice daily Lenabasum (*n* = 113) 5 mg orally twice daily Placebo (*n* = 115) Background immunosuppressive therapies (bIST) were allowed if doses were stable for ≥8 weeks before screening Duration: 52 weeks	Primary: ACR CRISS score—rCRISS Secondary: mRSS, HAQ-DI, FVC	Primary endpoint was not achieved Stable FVC in subjects treated with lenabasum 20 mg and MMF for more than 2 years compared to placebo	Less severe side effects in patients treated with lenabasum 20 mg compared to placebo

CRISS: American College of Rheumatology Combined Response Index in diffuse cutaneous Systemic Sclerosis (CRISS) score: MRSS, Health Assessment Questionnaire (HAQ) disability index (DI), physician global assessment of overall patient health, patient global assessment of health, FVC%.

**Table 3 biomedicines-10-00504-t003:** Head-to-head randomized double-blind clinical trials in SSc-ILD.

Molecule and Trial	Study Design	Endpoints	Efficacy	Safety
Rituximab vs. Cyclophosphamide (CTRI/2017/07/009152)	Randomization 1:1 RTX (*n* = 30): 1000 mg × 2 doses at 0, 15 days CYC (*n* = 30): pulses of CYC 500 mg/m^2^ Duration: 24 weeks	Primary: % predicted FVC Secondary: variables considered were: absolute change in liters (FVC-l); modified Rodnan skin scores, 6-min walk test, Medsgers score and new onset or worsening of existing pulmonary hypertension by echocardiographic criteria	A significantly higher percentage of patients experienced an improvement of FVC (%) in the RTX group vs. the CYC group (26.7% vs. 6.7%, respectively, *p* = 0.038). However, the rate of worsening of FVC (%) was similar in the RTX and CYC treated patients (3.3% vs. 3% *p* = 0.612)	The total number of patients having an adverse event was lower in the RTX group (30%) vs. the CYC group (70%) (*p* = 0.02) One patient developed severe pulmonary arterial hypertension 5 months after the completion of the trial (in the RTX group) and died.
Mycophenolate mofetil vs. cyclophophamide (NCT00883129)	Randomization 1:1 MMF (*n* = 69): 1500 mg orally twice daily for 104 weeks CYC (*n* = 73) 1.8 to 2·3 mg/kg orally once daily for 52 weeks + placebo for 52 weeks Duration: 104 weeks	Primary: % predicted FVC superior to CYC Secondary: the course from 3 to 24 months of the DLCO %-predicted, TDI and mRSS scores, and the change from baseline in quantitative HRCT scores for lung fibrosis and total ILD at 24 months	No significant difference between groups, with non-inferiority of the MMF arm vs. CYC arm.	Treatment related adverse events were more frequent in the CYC vs. MMF Time to withdrawal from the study medication or treatment failure was significantly shorter in the CYC arm. Sixteen deaths (11.3% of randomized patients) occurred during the 2-year course of the trial (11 CYC; 5 MMF)

## Data Availability

Not applicable.

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
