# Peer review of "Emerging Evidence and Treatment Perspectives from Randomized Clinical Trials in Systemic Sclerosis: Focus on Interstitial Lung Disease"

_biomedicines, 2022, doi:10.3390/biomedicines10020504_

Round 1

Reviewer 1 Report

The article is discussing results of recent studies and future possibilities. It is not offering practical advice for the practicing physician or specialist.  It is a good summary of the direction of current research efforts. It is useful to understand disease mechanism and treatment approach based on the understanding of disease mechanism.  There is no bias which therapy is superior to others, the authors do not promote molecular targets.  One could obtain up-to-date knowledge regarding emerging evidence and treatment perspectives for management of pulmonary involvement in systemic sclerosis.

English language usage is good.   

Author Response

We thank the reviewer for the comments.

Reviewer 2 Report

I have read the article by Aragona et al. with great interest. It is an interesting and detailed review article summarising clinical trials in SSc-ILD. I have only some, minor comments.

Comments:

  • “Nintedanib is an intracellular inhibitor targeting platelet-derived growth factor receptor, fibroblast growth factor receptor and vascular endothelial growth factor receptor tyrosine kinases recently approved for the treatment of idiopathic pulmonary fibrosis (IPF)[13].” Please, rephrase.
  • Please, describe the mechanism of action similarly to the other molecules.
  • Please, describe the mechanism of action similarly to the other molecules.
  • Figure 1. It would be important to describe if the vessel is part of the pulmonary or bronchial circulation.
  • Figure 1. In the middle of the figure there are 3 types of cells. Yet only, macrophages and neutrophils are named. What are the third cells? Eosinophils? Are they involved in the disease as well?

Author Response

 We thank the reviewer for the important comments. Please find below the answers:

Q1: “Nintedanib is an intracellular inhibitor targeting platelet-derived growth factor receptor, fibroblast growth factor receptor and vascular endothelial growth factor receptor tyrosine kinases recently approved for the treatment of idiopathic pulmonary fibrosis (IPF)[13].” Please, rephrase.

A1: We rephrased the quoted phrase in the manuscript

Q2: Please, describe the mechanism of action similarly to the other molecules.

A2: The missing mechanisms of action of the molecules discussed in the review have been added

Q3: Figure 1. It would be important to describe if the vessel is part of the pulmonary or bronchial circulation. 

A3: This is indeed a crucial point, but it appears outside the scope of the present review.  The possibility to draw the differences in the fibrotic process and highlight the early phase of vascular dearrangement is indeed a very intriguing topic and represents an important aspect of SSc-ILD pathogenesis. The figure itself has the main objective to schematically describe where each molecule, discussed in the text, acts. This might be the occasion to define a new topic of interest for future review article where each compartment of the scleroderma pathogenesis might be discussed in detail. Thanks for the very interesting suggestion.

Q4: Figure 1. In the middle of the figure there are 3 types of cells. Yet only, macrophages and neutrophils are named. What are the third cells? Eosinophils? Are they involved in the disease as well? 

A4: This is a very good observation. Indeed, additional granulocytes have been added in the figure, but as mentioned in the previous answer, we wanted to schematically depict essential targets for each molecule. In order to avoid any confusion in the reader, the additonal cells have been removed from the figure.